# Impact of the Food Matrix on the Antioxidant and Hypoglycemic Effects of Betalains from Red Prickly Pear Juice After In Vitro Digestion

**DOI:** 10.3390/foods14101757

**Published:** 2025-05-15

**Authors:** Roman-Maldonado Yvonne, Villanueva-Rodríguez Socorro Josefina, Mojica Luis Alfonso, Urías-Silvas Judith Esmeralda

**Affiliations:** Centro de Investigación y Asistencia en Tecnología y Diseño del Estado de Jalisco A.C. (CIATEJ), Guadalajara 44270, Jalisco, Mexico; yvroman_al@ciatej.edu.mx (R.-M.Y.); lmojica@ciatej.mx (M.L.A.); jurias@ciatej.mx (U.-S.J.E.)

**Keywords:** synergy, interactions, nutraceuticals, gastrointestinal stability, antidiabetic compounds

## Abstract

This study evaluated the impact of the food matrix on the bioaccessibility and hypoglycemic potential and antioxidant potential of betalains from red prickly pear juice (*Opuntia* spp.) after in vitro gastrointestinal digestion. Six aqueous model systems (AMSs) were formulated using a betalain extract combined with glucose, citric acid, mucilage, pectin, or all components, alongside three complex matrices, the fresh juice (FJ), a formulated beverage (BF), and a pasteurized formulated beverage (BP). In vitro digestion simulated the gastric and intestinal phases. The results showed that complex matrices (FJ, BF, and BP) enhanced betalain bioaccessibility, with FJ exhibiting the highest bioaccessibility (59%). Mucilage and pectin provided the strongest protection, reducing betalain degradation by 30% and 25%, respectively, while citric acid had a destabilizing effect. Pasteurization (BP) reduced betalain stability compared to FJ and BF. Antioxidant activity decreased post-digestion but remained higher in BF. Notably, FJ showed the highest inhibition of α-amylase (72%) and α-glucosidase (68%), surpassing acarbose (50–60% inhibition). These findings highlight the critical role of the food matrix, particularly mucilage and pectin, in stabilizing betalains through non-covalent interactions and enhancing their hypoglycemic potential. Red prickly pear juice emerges as a promising functional food for managing postprandial glucose levels, offering valuable insights for developing betalain-rich foods to address type 2 diabetes.

## 1. Introduction

Diabetes mellitus (DM) is a major public health challenge, affecting approximately 463 million people worldwide, with projections reaching 700 million by 2045 [1]. This metabolic disorder leads to prolongated hyperglycemia, which, if untreated, can result in severe complications such as neuropathy, nephropathy, retinopathy, and cardiovascular diseases [2,3]. Notably, 90% of people with diabetes have type 2 diabetes.

The primary goal of all diabetes treatment and management is to maintain an adequate blood glucose concentration. Currently, the available pharmacological drugs are expensive and have adverse effects, which include hypoglycemia, hepatotoxicity, and dyslipidemia. They often come with side effects and the risk of drug resistance [4,5]. Consequently, interest has grown in dietary approaches, particularly functional foods, as complementary strategies for diabetes prevention and management [6]. Some plant-derived bioactive compounds, such as polyphenols, have shown promising outcomes in inhibiting carbohydrate-digestive enzymes like α-glucosidase and α-amylase, thereby reducing postprandial glucose levels and improving insulin sensitivity [7,8,9].

Among bioactive compounds, betalains, a group of water-soluble nitrogen-containing pigments found in plants of the order Caryophyllales, have gained attention for their potential health benefits (antioxidant and anti-inflammatory properties) beyond their roles as natural colorants [10]. Studies suggest that betalains may contribute to glucose homeostasis in a specific pathway by enhancing the activity of glycolytic enzymes (glucokinase and pyruvate kinase) while inhibiting gluconeogenic enzymes (glucose-6-phosphatase and fructose-1,6-bisphosphatase), thereby reducing blood glucose levels [11,12,13]. Other works have studied betalains as a therapeutic approach for treating type 2 diabetes by lowering glucose absorption through the inhibition of intestinal carbohydrate-hydrolyzing enzymes (α-amylase and α-glucosidase) [14,15,16]. However, most studies evaluating these effects have used purified betalains or whole food matrices. This is a critical distinction because functional foods are complex systems where interactions between bioactive compounds and other food components can significantly alter their bioaccessibility [17].

A common assumption is that bioactive compounds within functional foods are readily bioaccesible and able to exert their beneficial effects; however, this is not always the case. The food matrix and digestive processes can modulate the bioaccessibility of these compounds, either enhancing their stability and absorption (synergistic effect) or reducing their effectiveness through interactions with other components [17,18]. The bioaccessibility and stability of betalains are influenced by the food matrix. Studies on beetroot juice have shown that processing techniques like high-pressure and supercritical carbon dioxide treatments can enhance their bioaccessibility. Additionally, the concentrations of betalains in the matrix affects their stability during digestion [19,20,21]. Therefore, evaluating the impact of the food matrix on betalain bioaccessibility—defined as the fraction that remains available for absorption after digestion—is essential for understanding their real potential in functional foods and their physiological impacts.

Red prickly pear (*Opuntia* spp.) is an excellent source of betalains alongside other bioactive components such as soluble fibers, polyphenols, and flavonoids, which may contribute to betalain stability and bioaccessibility [15,22,23]. The physicochemical properties of betalains, including their charge distribution and potential interactions with other food components, can affect their release and absorption during digestion. Additionally, their efficacy in modulating carbohydrate metabolism at the gastrointestinal level by inhibiting α-amylase and α-glucosidase may vary depending on these interactions [13,24,25].

This study aimed to evaluate the impacts of food matrix components of red prickly pear fruit on betalain bioaccessibility and carbohydrate-degrading enzyme inhibition potential after in vitro digestion. Aqueous model systems were formulated using betalains and key juice components (pectin, mucilage, citric acid, and glucose), along with complex food matrices, including fresh red prickly pear juice (reference sample) and a formulated beverage (fresh and pasteurized), to assess the influences of matrix complexity and thermal processing on betalain bioaccessibility and functionality. These findings provide insights into optimizing functional foods with betalains or red prickly pear fruit juice as ingredients for blood glucose level control.

## 2. Materials and Methods

### 2.1. Raw Material

Red prickly pear fruits belonging to the *Opuntia* genus, “Zacatecas” variety (*Opuntia* spp.), were obtained from a local market in Jalisco, Mexico, and came from the state of Zacatecas, Mexico. Fruits selected for the study were free from mechanical or microbiological damage on the skin and were at a stage of commercial ripeness. They were washed with water and soap and stored at 5 °C until juice was obtained. For juice obtention, the red prickly pear fruits were peeled, the seeds were separated from the peel using a strainer, and the juice was frozen until use.

### 2.2. Betalain Extract Obtention from Red Prickly Pear Juice

#### 2.2.1. Betalain Extraction

In preliminary tests, seven different previously reported methods were evaluated. The methods varied in solvent polarity, extraction time, the presence/absence of light, purification method, and times. The method reported by Cervantes-Arista et al. [26] was selected, considering the amount of betalains and the polar green solvent used. To obtain a betalain extract, 150 g of the fresh Zacatecas variety of red prickly pear juice was used, 675 mL of the solvent (distilled water) was added and stirred for 2 min in a dark room, and then centrifuged at 16,000× *g* for 5 min, and the supernatant was collected. The supernatant was frozen at −20 °C for 72 h for subsequent lyophilization, aiming to concentrate and prevent the degradation of betalains.

#### 2.2.2. Purification of the Betalain Extract by Gel Permeation Chromatography (GPC)

Considering that due to the use of water as a polar solvent to obtain the betalain extract (BE), the extract also contained sugars, so, gel permeation chromatography (GPC) was used to remove these sugars following the method reported by Gonçalves et al. [27]. The Bx of each fraction obtained was determined and the identity of residual sugars was determined using thin-layer chromatography (TLC).

Finally, the fractions obtained were lyophilized. The content of betalains was measured using a spectrophotometric method (Thermo Scientific™ Multiskan™ GO Microplate, Waltham, MA, USA), and the UV/vis absorption spectra were recorded from 200 to 700 nm to obtain absorption values at their respective absorption maxima. Measurements were performed in triplicate, and the betalain content (BC) was calculated according to the method reported by Gonçalves et al. [27] (Equation (1)):Betalain content [mg/g] = [(A(DF)(MW) V) × 1000/(εLPL)](1)
where A is the maximum absorption values at 538 and 490 nm for betacyanins and betaxanthins, respectively, DF is the dilution factor, V is the volume of initially extracted solution (in liters), LP is the lyophilized pulp weight (g), and L is the path length (0.64 cm) of the plate. The molecular weight (MW) and molar extinction coefficient (ε) of betanin [MW= 550 g/mol; ε = 60,000 L/(mol cm) in H_2_O] were applied to quantify the betacyanins, and quantitative equivalents of the major betaxanthins (Bx) were determined by applying the mean molar extinction coefficient [ε = 48,000 L/(mol cm) in H_2_O].

### 2.3. Aqueous Model System Formulation

Different aqueous model systems (AMSs) were formulated (S1–S6) to simulate the complexity of red prickly pear juice, containing a fixed concentration of the betalain extract (BE). The most abundant components in the prickly pear juice ranged from small molecules such as citric acid and glucose to more complex compounds such as polysaccharides (mucilage and low-methoxyl pectin). The exact amount of each component was previously determined in the juice. The aqueous model systems (AMSs) were formulated by combining the betalain extract (BE) with each component individually (glucose, citric acid, low methoxyl pectin, and mucilage) in water (S1–S5), as well as by creating a final, more complex system that closely resembles the composition of red prickly pear juice (S6). The concentrations of each component were established based on their content per 100 g of fresh red prickly pear juice (JF).

Additionally, two more complex model systems were formulated. The first formulated beverage (BF) (patented formulation) was based on red prickly pear juice, and the second was the same drink subjected to a conventional pasteurization process following the method described by Ayala Bendezú [28] to obtain the pasteurized beverage (BP).

Table 1 summarizes and represents the formulated aqueous model systems, their components, chemical features, and their classification based on complexity and the three complex food matrices used: the fresh juice (JF), the formulated beverage (BF), and the formulated beverage pasteurized (BP).

Subsequently, the antioxidant potential of the samples was determined by two different methods (ABTS and DPPH assays) to understand the effect of the interactions of betalains and components of the food matrix and their impacts on betalain bioaccessibility.

### 2.4. Antioxidant Potential

#### 2.4.1. DPPH Assay

DPPH (1,1-diphenyl-2-picrylhydrazyl) assay was carried out according to the method reported by Virgen-Carrillo et al. [29], with some modifications. The AMS and the complex food matrices were diluted to a 1:4 ratio with distilled water (0.075 mg/mL). In a 96-well flat-bottom plate, 20 μL of the diluted samples, Trolox standard (0.1–1.5 mM), or blank (distilled water) were loaded, followed by the addition of 180 μL of 0.06 mM DPPH in 80% methanol, and then the plates were incubated in the dark for 30 min. After the incubation, the absorbance was read at 517 nm using a spectrophotometer (Thermo Scientific™ Multiskan™ GO Microplate, Waltham, MA, USA), and the results are expressed as micromoles of Trolox equivalents.

#### 2.4.2. ABTS Assay

The ABTS (2,2-azinobis-3-ethylbenzothiazoline-6-sulphonic acid) assay was performed according to the method reported by Virgen-Carrillo et al. [29], with some modifications. Samples were diluted to a 1:4 ratio with distilled water (0.075 mg/mL). In a 96-well flat-bottom plate, 20 μL of the diluted samples, Trolox standard (0.05–0.75 mM), or blank (deionized water) were added, followed by the addition of 180 μL of 7 mM ABTS^+^ radical solution. The absorbance was read at 734 nm using a spectrophotometer (Thermo Scientific™ Multiskan™ GO Microplate, Waltham, MA, USA). Trolox was used as a standard for the calibration curve, and the results are expressed as micromoles of Trolox equivalents.

### 2.5. In Vitro Digestion

The six aqueous model systems, the betalain extract (BE), the two drinks, and the fresh juice were subjected to in vitro digestion following the methodology proposed by Domínguez-Murillo and Urías-Silvas [30], with some modifications.

Gastric stage: One milliliter of each sample was taken and mixed with 90 mL of a 2% *w*/*v* NaCl electrolyte solution; 3.2 g/L of porcine pepsin enzyme *w*/*v* [≥250 units mg^−1^] (EC. 3.4.23.1, Sigma-Aldrich, Naucalpan de Juarez, Mexico) was also solubilized at 25,000 U ml^−1^, the pH of each sample was adjusted at pH 2.0 with HCl (1 N), and the samples were left under constant stirring (100 rpm) in a water bath at 37 °C for 2 h. One milliliter of the samples submitted to the gastric phase was saved for subsequent analysis; the remaining volume was used to continue with the intestinal stage.

Intestinal stage: The pH of the samples was adjusted to 7.0 with NaOH (1 N), and then 1 mL of an aqueous solution of pancreatin enzyme [8× UPS] (EC. 232-468-9, Sigma-Aldrich, Naucalpan de Juarez, Mexico) (0.2%) *w*/*v* and 1 mL of an aqueous solution of bile salts at 3% *w*/*v* were added to the remaining volume of the previous phase. The digestion was incubated for 4 h in a shaking water bath (100 rpm) at 37 °C.

Bioaccessibility was calculated based on the concentration of betalains determined by spectrophotometry. The relationship between the concentration of betalains in the intestinal aqueous phase at the end of digestion (supernatant) and their initial concentration in the aqueous model system (AMS) were obtained (Equation (2)).Bioaccessibility = [“Betalain content in the supernatant”/“Betalain content in the AMS”] × 100(2)

The final volume was further subjected to enzymatic inhibition tests using two digestive enzymes in the small intestines.

### 2.6. Enzymatic Inhibition Assay

For this stage, all aqueous model systems were evaluated; also, the components of each model system (without betalains) were used as comparison to find out if any of them individually had an inhibitory effect on the activity of the enzyme or if they promoted synergy with the betalains. The drug acarbose (1 mM) in two different presentations, Pharmalife^®^ drug grade and Sigma^®^ A8980-1G reagent grade, was used as a positive control. This procedure was performed following the methodology of Gómez-Maqueo and García-Cayuela; Mojica et al.; Oluwagunwa et al. [15,31,32], and the Merck method for the enzyme α-amylase [33], with modifications.

#### 2.6.1. α-Amylase Inhibition

Samples and acarbose (positive control) or PBS (negative control) were prepared in water. The inhibitory effect on α-amylase was determined by quantifying reducing sugar contents using dinitrosalicylic acid (DNS). Samples or controls of 100 μL of (0.3 mg/mL of betalains) in AMS and the complex food matrices were added to 100 μL of α-amylase (1 U/mL) and 200 μL of sodium phosphate buffer (20 mM, pH 6.9) to obtain a 0.075 mg/mL final concentration of betalains. The samples were pre-incubated at 25 °C for 10 min, and 200 μL of 1% starch prepared in 20 mM sodium phosphate buffer (pH 6.9) was added. The reaction mixtures were incubated at 25 °C for 10 min. The reactions were stopped by incubating the mixture in a boiling water bath for 5 min after adding 1 mL of dinitrosalicylic acid. The reaction mixtures were cooled to room temperature and diluted with distilled water to reach a final volume of 12 mL. The amount of reducing sugars released by its enzymatic action was determined spectrophotometrically (Thermoscientific^TM^ multiscan^TM^ Go, USA) using a maltose standard curve (0.5 to 2 mg/mL) at 540 nm in a 96 multiwell plate with 300 µL of each sample. A control and blank sample were used to subtract the betalain interference at 540 nm. This procedure was also carried out for all samples with and without enzyme to subtract the noise value from the samples with enzyme and have the net values of enzyme inhibition. The values were obtained by these three independent experiments. The enzymatic inhibition percentage was calculated as shown in Equation (3) and reported by Oluwagunwa [32]:enzymatic inhibition = (AbsControl − (AbsSample − AbsSaBco))/AbsControl × 100(3)
where

AbsControl is the maximum absorbance of starch without the enzyme (C -);AbsSample is the net absorbance of each sample;AbsSaBco is the absorbance of each sample without enzyme.

#### 2.6.2. α-Glucosidase Inhibition

For the α-glucosidase inhibition assay, the method reported by Virgen-Carrillo et al. [29] was used with slight modifications. The samples were diluted at a 1:8 ratio, equivalent to 0.07 mg/mL of betalains in distilled water. Then, 50 μL of each diluted sample, acarbose (positive control), or PBS (negative control) was added to a 96-well plate, followed by 100 μL of α-glucosidase enzyme (1 U ml^−1^ from *Saccharomyces cerevisiae* PBS, 0.1 M pH 6.9), and incubated for 10 min at 37 °C with stirring intervals (10 s); after 10 min, the absorbance was measured at 405 nm in a spectrophotometer (Thermoscientific^TM^ multiscan^TM^ Go, USA) to subtract the betalain interference. Then, 50 μL of the substrate 4-nitrophenyl α-D-glucopyranoside (4p-NPG [5 mM] in PBS [0.1 M], pH 6.9) was added to each well and incubated at 37 °C for 5 min. Finally, the absorbance was measured at 405 nm, which represents the 4-nitrophenol released by enzymatic activity. The results are expressed as α-glucosidase inhibition percentage (Equation (3)).

#### 2.6.3. Statistical Analysis of the Data

Statistical analyses were conducted using R (R Core Team, Vienna, Austria, 2024). Data visualization was performed using the ggplot2 package [34], while data manipulation was carried out with the dplyr package [35]. Post hoc comparisons were visualized using the multcompView package [36]. The data are presented as the means and standard deviations of three replications. Tukey’s HSD test was used to detect significant differences between treatments. A significance level of *p* < 0.05 was considered for all statistical tests.

## 3. Results and Discussion

### 3.1. Extraction of Betalains from Red Prickly Pear Juice

Table 2 presents the yield of the betalain extract obtained from fresh red prickly pear juice (Zacatecas variety). A total of 39.6 mg of betalains were present in 150 g of fresh juice. Using water as solvent, 2.224 mg betalains/g juice was obtained. Starting with 150 g of juice resulted in a total extraction of 33.36 mg of betalains (Table 2). Based on this information, the calculated extraction rate was 84.24%, reflecting the significant efficiency of the extraction process. The purification of the extract was achieved through gel permeation chromatography (GPC), a technique that effectively eliminated sugars from the sample. This purification process significantly improved the extract’s quality by minimizing impurities and increasing the concentration of betalains. The extract was used to simulate a food matrix, preserving the full diversity of betalains in red prickly pear juice.

### 3.2. Antioxidant Potential of Betalains and AMSs

Figure 1 shows the antioxidant potential of AMSs formulated with the betalain extracts and the complex food matrices FJ, BF, and BP against two different radicals (DPPH and ABTS) before and after the digestion process. A decrease in the antioxidant potential of all samples against the ABTS radical was observed after digestion. Before digestion, the complex model systems presented a lower scavenging potential against the DPPH compared to the ABTS radical (Figure 1a,b). Before digestion, the sample with the highest antioxidant activity was BE (betalain extract), followed by the model systems (S1–S5), verifying that the bioactive compounds of the prickly pear juice are betalains and their interactions with the components of the juice matrix affected and reduced the amount of free betalains and, therefore, their antioxidant potential was lower. After digestion, the sample with the lowest antioxidant potential was BE (betalain extract), which also confirmed that the components of the food matrix were necessary to protect betalains during the digestion process.

Comparing the results of the fresh (BF) and pasteurized beverage (BP), it was observed that the BF had a lower antioxidant potential compared to BP. This could be due to pasteurization that degrades complex polysaccharides, allowing the release of betalains and improving their interaction with the free radicals. This tendency was observed before and after digestion. Although specific data on betalains’ thermal stability in red prickly pear juice are limited, existing studies highlight key factors that contribute to their preservation during pasteurization. Herbach et al., in 2006 [37], mention that citric acid acts as a chelating agent, helping to stabilize betalains by binding metal ions that would otherwise catalyze their degradation. Additionally, thermal treatment inactivates enzymes such as polyphenol oxidases and peroxidases, which are known to degrade betalains, further contributing to their stability. Moreover, the matrix composition of the juice, including sugars and organic acids, likely plays a protective role, preventing betalain isomerization during heat processing [38].

It has been reported that the common mechanisms by which betalains reduce free radicals involve a single proton transfer followed by electron transfer (SPLET) or hydrogen atom transfer/proton-coupled electron transfer (HAT/PCET). The difference between both is that HAT/PCET could be the possible mechanism due to the low bond dissociation energy (BDE) required and their charge stability in aqueous media [27,39]. The values of the antioxidant potential are shown in Table A2. To scavenge the ABTS^+^ radical, the transfer of hydrogen atoms by the betalains is necessary. Therefore, in S1 (betalains + water), the phenolic group is possibly not conjugated with the diazapolymethine system and is free to perform its function (Figure 2). Still, when there are more components in the aqueous model systems, the formation of interactions that stabilize betalains occurred, as observed in samples FJ (fresh juice), BF (fresh beverage), and BP (pasteurized beverage), thus limiting the free functional groups of betalains that could perform the antioxidant function of the betalains by scavenging the radical.

Conversely, evaluating the antioxidant potential of the samples against the DPPH radical revealed a progressive increase with the growing complexity of the model system. This is because the DPPH radical scavenging mechanism is through the transference of electrons by the conjugated double bonds within the aromatic rings, and so an ionization potential (IP) and energy transfer enthalpy (ETE) are required and more significant than those required to give up an H^+^ atom; these results corroborate that the primary mechanism by which betalains reduce free radicals is through the transfer of hydrogen atoms. However, when betalains interact with other food matrix components, this mechanism may be hindered, allowing betalains to transfer electrons instead of releasing an H^+^, even though this process requires more energy; this is possible due to their diazapolymethine system (Figure 2) [27]. It can be inferred from systems S4 (BE + pectin) and S5 (BE + mucilage), where betalains are combined with hydrophilic polysaccharides, that they can interrelate through dipole–dipole interactions, hydrogen bonding, and electrostatic interactions due to the slightly acidic pH of the aqueous model systems. Additionally, in more complex systems such as S6 (BE + all the components), JF (fresh juice), and beverages (BF and BP), betalains, being polar nitrogen compounds, can interact well with food matrix components such as carbohydrates (e.g., mucilage, pectin, and dietary fiber), proteins, and other small molecules through weak non-covalent forces (e.g., hydrogen and van der Waals bonds). These complex interactions may contribute to the enhanced antioxidant potential observed in these systems. Although JF, BF, and BP were not formulated in this study, the model systems (S1–S6) were designed to preserve the betalain content comparable to that of natural matrices.

Other studies have explored the role of the structure in the antioxidant potential of betalains and betalamic acid, concluding that phenolic betalains exhibit a higher antioxidant capacity compared to non-phenolic derivatives [12,40,41,42,43]. However, the knowledge of structure–property relationships of betalains is still limited compared to other important classes of plant pigments, such as polyphenols, anthocyanins, and carotenoids.

The antioxidant potential of betalains is attributed to their unique structure (Figure 2), which includes phenolic groups and conjugated systems that facilitate electron or proton donation, depending on the interactions occurring within the food matrix, enabling them to neutralize free radicals effectively [27,44,45]. As a result, betalains demonstrate promising applications in treating oxidative stress. These results highlight the significant antiradical capacity of betalains, and such interactions might also hinder betalains from performing other crucial biochemical functions in the body [27,46,47].

The antioxidant activity of betalains serves not only as a marker of their reducing capacity but also as an indicator of their structural integrity. This structural stability is essential for their function as enzyme inhibitors. For example, Gandía-Herrero et al. (2012) describe that betalains with a high antioxidant potential (those with a high content of -OH groups in the C6 position) and an electrophilic structure are more effective at inhibiting digestive enzymes [41]. Additionally, Slimen et al. (2017) reported that betalains, due to their amine group, may participate in various biological processes, thanks to their hydrogen bonding properties [45]. Therefore, the preservation of their structure not only helps in antioxidant defense but also allows them to maintain the ability to interact with digestive enzymes, potentially influencing blood sugar regulation.

### 3.3. Betalain Bioaccessibility and Hypoglycemic Effect After In Vitro Digestion

#### 3.3.1. In Vitro Digestion Bioaccessibility

Figure 3 shows the total contents of betalains in each model system and their percentage of bioaccessibility after in vitro digestion The results show that the amount of free betalains in the evaluated samples varied according to the ingredient with which they were mixed, the extract containing seven times more betalains (1.3 mg/g) than the formulated model systems (S1–S6), which had values ranging between 0.30 and 0.32 mg/g. These values are consistent with those reported in the literature for red prickly pear pulp and juice, which range from 0.11 to 0.60 mg/g (fresh weight) [24,48,49]. In this work, model systems were designed with approximately 0.3 mg/g of betalains, and the complex systems such as FJ, BF, and BP contained between 0.2 and 0.5 mg/g of betalains, representing the average betalain content in red prickly pear fruit and highlighting the protective effect of the juice matrix on the remaining betalains after in vitro digestion. Fresh juice (FJ) retained the highest amount of total betalains after digestion, followed by system S6, which mimics the complexity of juice with its most representative components.

During digestion, the gastric phase was the most critical for betalain degradation, as this phase caused a significant reduction in the betalain concentrations. This was due to the low pH produced by gastric acids, which promoted the protonation and destabilization of betalains, rendering them highly susceptible to degradation [50]. For instance, the BE, despite having the highest initial concentration of betalains (1.3 mg/g), exhibited a marked reduction (87%, 0.18 mg/g) during the gastric phase when compared to the other systems and complex matrices. This observation highlights that betalains do not inherently possess resistance to digestive conditions, and their bioaccessibility mainly depends on the composition and structure of the food matrix in which they are embedded. It also confirms that bioaccessibility is more relevant than the initial concentration of the compound, as interactions with the matrix can either limit or enhance the stability and final biological activity of betalains [24,51].

During the gastric phase, the BP (pasteurized beverage), S2 (BE + glucose), and S5 (BE + mucilage) systems exhibited the highest degradation rates (62%, 46%, and 42%, respectively). In the case of the pasteurized beverage (BP), the high degradation rate could be attributed to the heating process (80 °C) inherent to the pasteurization, which may promote the degradation of both betalains and polysaccharides (S2 and S4); in particular, heating can cause chain fragmentation through the cleavage of α-(1→4) linkages or the removal of methyl groups, reducing their stability and size during digestion [52].

Fresh juice (FJ) exhibited a higher percentage of bioaccessibility (59%), followed by the beverage formulated (BF) (58%), equivalent to 0.1–0.2 mg/g of remaining betalains. This suggests that betalains require other components, specifically, sugar-rich compounds such as polysaccharides (e.g., pectin and mucilage), to stabilize their structure during digestion. For instance, systems S4 (BE + pectin), S5 (BE + mucilage), and S6 (BE + citric acid, glucose, pectin, and mucilage) helped maintain the integrity of the betalains’ conjugated double bond system [45,53,54]. Additionally, the glucose in the solution may interact with free protons (H^+^), reducing local protonation activity around the betalain molecules and thereby slowing down their degradation at sensitive functional sites [50].

All the bioaccessibility percentages and betalain contents during in vitro digestion are presented in Table A1.

The S5 system, which contained the betalain extract and mucilage, showed one of the lowest betalain reduction rates. It is suggested that mucilage may interact with betalains, stabilizing them and allowing them to reach the intestinal phase, where they must be absorbed to carry out specific functions in the body. The main functional groups in betalains that interact with other components are carboxyl or hydroxyl groups, specifically carbons C10, C19, and C20 [51,55,56]. Additionally, the S4 system, which contains pectin, another abundant polysaccharide in red prickly pear juice, exhibited a higher betalain content after the gastric phase than the other systems. This could be due to pectin’s low degree of methoxylation, which may favor the exposure of free polar regions (COO^-^) capable of interacting with betalains, promoting the formation of esterified betalains and protecting them from oxidation during digestion [50].

Systems containing pectin (S4) and mucilage (S5) exhibited higher remaining betalain contents. It is important to note that while these systems retained higher amounts of betalains, the percentage of bioaccessibility varied.

A comparison between betacyanins and betaxanthins showed that betacyanins predominated in red prickly pear juice, even after the gastric phase. However, betaxanthins became more prevalent by the end of the digestion process. This could be attributed to the degradation of betalains into colorless structures or their precursors, such as cyclo-DOPA and betalamic acid, reducing their chromophore capacity [50].

The results obtained in this study align with those reported in other works that also studied the effect of the food matrix on betalains’ bioaccessibility. However, the bioaccessibility of betalains in red prickly pear juice (58%) was higher than reported beetroot and cactus berry fruits [24,48]. The food matrix components and their interactions with betalains prevented their degradation in the gastrointestinal tract until they reached their active site. The protection level provided by the food matrix is an important factor to consider. It could be possible that the juice matrix protects betalains during the digestion process, increasing the amount of betalains that finally reaches the intestine so that they can perform other specific functions, such as inhibiting digestive enzymes. This study found that as the complexity of the food matrix increases, the bioaccessibility of betalains also increases. At the same time, the antioxidant activity decreases due to the interactions between betalains and the food matrix. Complex matrices can confer protection during digestion, thereby promoting bioaccessibility by mitigating the degradation of compounds. Nevertheless, excessive protection could potentially hinder the release and subsequent absorption of betalains in the small intestine. So, achieving an optimal balance between enhancing stability throughout digestion and ensuring effective bioaccessibility is necessary.

#### 3.3.2. Hypoglycemic Effect

##### α-Amylase Inhibition

α-Amylase is an enzyme responsible for breaking down dietary starch into oligosaccharides and then into monosaccharides that can be absorbed by epithelial cells [32]. Therefore, its inhibition is considered an active strategy to reduce the amount of glucose available, which can help manage type 2 diabetes. The inhibitory potential of AMSs (aqueous model systems) was compared with the acarbose, which is responsible for lengthening duration of the carbohydrate absorption and hence reducing plasma glucose levels over time [8].

Figure 4 shows the potential inhibitory effects of all model systems before and after in vitro digestion. The components of the model systems were individually assessed for their potential inhibitory effects. Although ingredients without betalains had an inhibitory effect on enzyme activity, this was significantly lower than when combined with betalains. However, it is important to notice that P (pectin), M (mucilage), CA (citric acid), and G (glucose) inhibited α-amylase activity after digestion; this could be because P and M are soluble fibers that create steric hindrance between the enzyme and its active site during digestion, inducing changes in the enzyme conformation. Also, the enzymes are sensitive to pH, and so the presence of CA can influence its activity by altering the ionization state of key amino acid residues in both the active site and other parts of the enzyme structure that affect its activity [2].

The AMSs (aqueous model systems), both beverages (formulated and pasteurized beverages), as well the betalain extract (BE) inhibited between 87% and 98% of α-amylase activity; the S1 (BE + water) and S5 (BE + mucilage) were the most potent models to inhibit α-amylase even after digestion and compared to acarbose in its two presentations: pharmaceutical (AP) and reagent grade (AR). Moreover, AP increased its inhibitory potential after in vitro digestion, which is associated with the formulation requirements of medications to ensure their efficacy through the digestive process (excipient), in contrast to the inhibitory capacity of AR, which diminished after digestion.

Given the low inhibitory effect observed for the ingredient solutions without betalains, it is clear that betalains play a crucial role in enhancing the inhibitory effect. This was demonstrated by the significant inhibition achieved with both AMSs and JF (fresh juice) and was evident an increase in the inhibition of α-amylase activity was evident compared to systems without betalains. These results suggested that the hypoglycemic effect is primarily attributable to the presence of betalains. A potential interaction between betalains and other components in the model systems may be responsible for this effect. However, it was evident that a greater complexity of the matrix (S6, FJ, and BP) tends to reduce the inhibitory effect on the enzyme.

To date, limited information is available to discuss the precise mechanism of action of betalains to block this enzyme. α-Amylase consists of two regulation sites (Ca^+^ binding and Cl) responsible for maintaining its structural integrity and could be related to the activation on enzymatic function [2,57]. The active site consists of three amino acid residues (Asp 197, Asp 300, and Glu233), where it has been reported to have a strong interaction with other bioactive compounds (mainly polyphenols), reducing its enzymatic efficiency [2,29,32].

Considering previous studies on polyphenols such as ellagic acid and epicatechin [29,32], which share structural similarities with betalains, particularly their double bond system, aromatic rings, and some polar regions, it can be postulated that betalains inhibit α-amylase activity through a competitive mechanism. This inhibition likely occurs due to the affinity of the catalytic site of betalains for the other compounds with similar chemical structures, such as benzene sulfonamides [58]. Their polarity favors the formation of hydrogen bonds and Van der Waals forces, promoting stable binding, distorting the substrate, and reducing enzymatic efficiency. Additionally, in prickly pear juice (FJ), the components of the food matrix might enhance this effect by inducing changes at the enzyme’s regulatory sites, thereby promoting the inhibition and stabilizing betalains during digestion and allowing them to exhibit a stronger inhibitory effect on α-amylase compared to recognized inhibitors such as acarbose.

##### α-Glucosidase Inhibition

α-Glucosidase is a naturally occurring enzyme in the intestine that is responsible for releasing glucose molecules from disaccharides previously broken down by enzymes such as α-amylase, which cleaves α-glycosidic bonds. As shown in Figure 5, acarbose (drug known for its hypoglycemic effect through α-glucosidase inhibition) exhibited lower inhibitory activity against this enzyme (13% and 20% after digestion) compared to its effect on α-amylase and to the inhibition observed in the betalain-containing model systems. These results are consistent with those reported by Virgen-Carrillo et al. [29]. Additionally, a significant decrease in the inhibitory capacity of all samples was observed after in vitro digestion.

However, BE has the greatest inhibitory effect on α-glucosidase, followed by systems S2, S6, and JF that, in addition to betalains, have 13.7% of glucose (average amount present in fresh juice). This indicates that there could be inhibition by product formulation but does not inherently lead to a decrease in blood sugar levels that is because other mechanisms, such as the presence of free monosaccharides or alternative enzymatic pathways can compensate for the partial inhibition of α-glucosidase. Thus, while elevated glucose levels in the intestine may suppress α-glucosidase activity, the overall effect on blood sugar regulation is influenced by the availability of alternative sugar sources and the efficiency of compensatory absorption pathways.

Also, the inhibition byproduct occurs when the accumulation of glucose, the enzyme’s reaction product, slows or inhibits further enzymatic activity. Specifically, glucose competes with the substrate for binding at the enzyme’s active site, reducing its ability to process additional substrates. This suggests that, within the context of digestion, excess glucose may naturally limit α-glucosidase activity as a regulatory mechanism to prevent an overload of glucose entering the bloodstream [59]. As previously reported, betalains exhibit hypoglycemic properties primarily through their ability to inhibit carbohydrate-digesting enzymes such as α-glucosidase and α-amylase. Several studies have demonstrated that betalain-rich extracts from Beta vulgaris, *Opuntia* ficus-indica, and other sources significantly reduce α-glucosidase activity in vitro, supporting their potential as natural antidiabetic agents [12,60,61,62]. However, the exact mechanism by which betalains exert this inhibitory effect remains uncertain. Based on our findings, we hypothesize that betalains may act as competitive inhibitors by structurally mimicking disaccharide substrates, allowing them to bind to the active site of α-glucosidase and prevent the access of natural substrates such as maltose or sucrose. Therefore, systems containing betalains and glucose may exhibit the highest effect on inhibiting this enzyme through competitive inhibition. For instance, the S2 system (BE + glucose), as well as the more complex systems e.g., fresh juice (JF), formulated beverage (BF), and pasteurized beverage (BP), where glucose is predominant, could exhibit an enhanced inhibitory capacity of betalains against the enzymes. This is likely due to a higher predisposition to form glycosylated structures. As often observed for phenols, glycosylation drastically increases the stability of the compounds, mainly by impeding their oxidative degradation by blocking phenolic groups [53,63]. In nature, many natural products, especially the secondary metabolites of microbes and plants, are glycosylated, which changes metabolites’ bioactivity and physiochemical properties. The addition of a sugar moiety increases polarity and solubility and often results in improved stability of natural products [64].

The inhibition by the betalain extract (BE) could be through the competition inhibition of glucose molecules by betalains, as reported for other bioactive compounds [65]. Also, it has been reported that van der Waals and hydrogen bonds are common bonds observed in all interactions between some bioactive compounds (p-coumaric acid, vanillin, hexahydrofarnesyl acetone, and phytol) and α-amylase and α-glucosidase. These changes observed in the bonds formed during interactions can be associated with the different amino acid contents of each enzyme and its accessibility. As a result, these interactions of betalains with the enzymes can cause changes in the three-dimensional structures and a loss of function, triggering their inhibition [66].

## 4. Conclusions

This study demonstrates the significant impacts of the food matrix on the bioaccessibility, antioxidant potential, and hypoglycemic effect of betalains from red prickly pear juice. The retention of antioxidant activity after in vitro digestion suggests that betalains may preserve key structural features necessary for their radical-scavenging capacity due to the protective effect of the food matrix. This observation supports the hypothesis that, despite partial degradation, betalains, when associated with other food components in the fresh juice, retain sufficient molecular integrity to continue exerting beneficial effects after gastrointestinal digestion. The presence of mucilage and pectin plays an important role in the food matrix because, despite being polysaccharides with polar regions and interacting with betalains, they protect the structure of betalains throughout digestion. This protective effect is attributed to non-covalent interactions between betalains and matrix components, such as polysaccharides (mucilage and pectin), stabilizing betalains’ structure and mitigating their oxidative degradation. Moreover, betalains exhibited significant inhibitory effects on carbohydrate-digesting enzymes, α-amylase and α-glucosidase, with fresh juice (FJ) showing the highest inhibition rates (72% for α-amylase and 68% for α-glucosidase). This suggests that red prickly pear juice could be a natural alternative for managing postprandial glucose levels, potentially complementing dietary strategies for type 2 diabetes management. Red prickly pear juice, with its natural composition of betalains, mucilage, and pectin, emerges as a promising candidate low-cost, sustainable, functional food, especially in arid and semi-arid regions where *Opuntia* species thrive with minimal water requirements. Future research should focus on evaluating the hypoglycemic effects of red prickly pear juice in human trials, investigating processing techniques that minimize betalains’ degradation while maintaining their bioactivity, and exploring the molecular interactions between betalains and matrix components to understand their stabilization and inhibitory mechanisms better.

## Figures and Tables

**Figure 1 foods-14-01757-f001:**
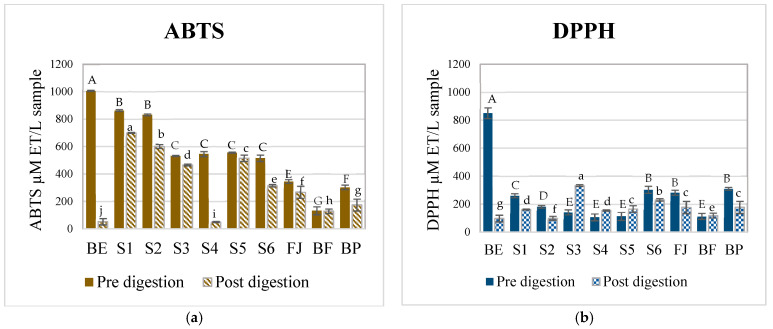
Antioxidant potential of different complex aqueous model systems before and after digestion. (**a**) ABTS assay. (**b**) DPPH assay. BE, betalain extract, S1, BE in water; S2, BE + glucose; S3, BE + citric acid; S4, BE + pectin: S5, BE + mucilage, S6, BE + glucose, citric acid, pectin, and mucilage; JF, fresh juice; BF, formulated beverage; and BP, pasteurized beverage. The data are presented as the means ± SDs from three independent replicates. Uppercase letters indicate statistically significant differences (*p* < 0.05) between antioxidant activities (DPPH and ABTS) before digestion, while lowercase letters indicate differences after digestion (*p* < 0.05) according to Tukey’s test.

**Figure 2 foods-14-01757-f002:**
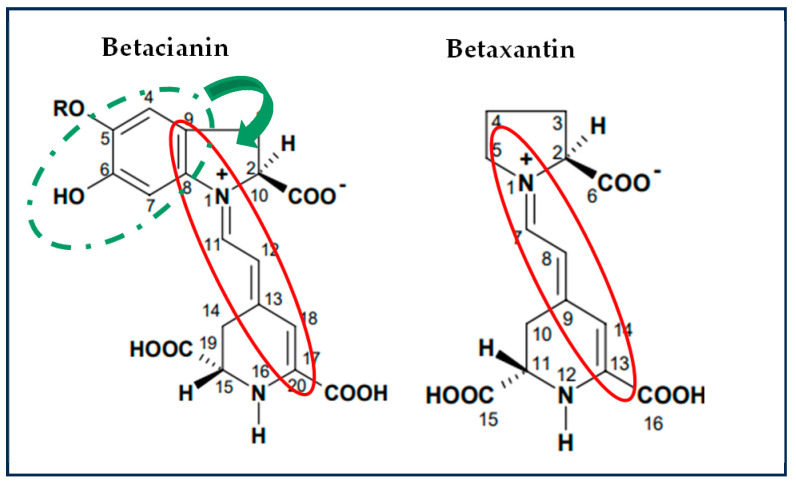
Chemical structures of betalains and their possible functional groups involved in radical reduction. Red circle: diazapolymethine system. Dotted green circle: phenol group. *Adapted from “Estudio preliminar de los pigmentos presentes en cáscara de pitaya de la región mixteca” by Mandujano, 2006 Tesis de licenciatura, maestría, Universidad Tecnológica de la Mixteca.*

**Figure 3 foods-14-01757-f003:**
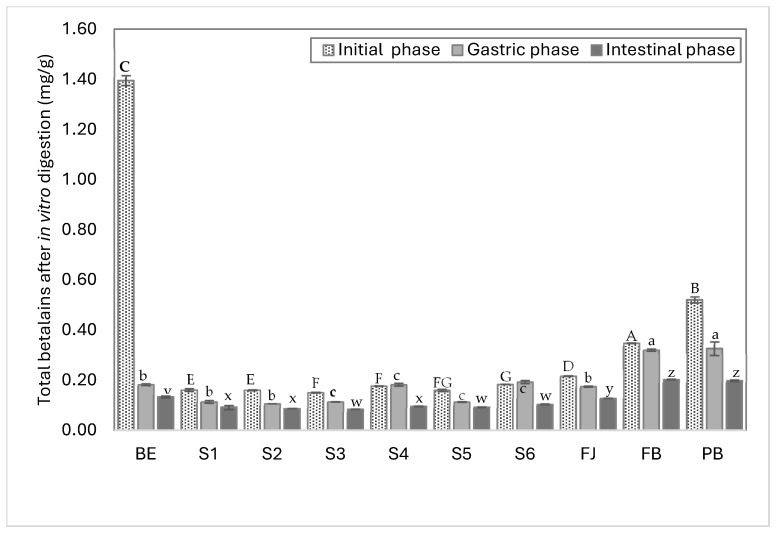
Free betalain contents in aqueous model systems during in vitro digestion quantified by spectrophotometry UV/vis. BE, betalain extract; S1, BE in water; S2, BE + glucose; S3, BE + citric acid; S4, BE + pectin; S5, BE + mucilage; FJ, fresh juice; FB, formulated beverage; BP, pasteurized beverage. Uppercase letters indicate statistically significant differences between samples in the initial digestion phase, lowercase letters indicate differences in the gastric phase, and lowercase letters from the end of the alphabet indicate differences between samples in the intestinal phase (*p* < 0.05) according to Tukey’s test.

**Figure 4 foods-14-01757-f004:**
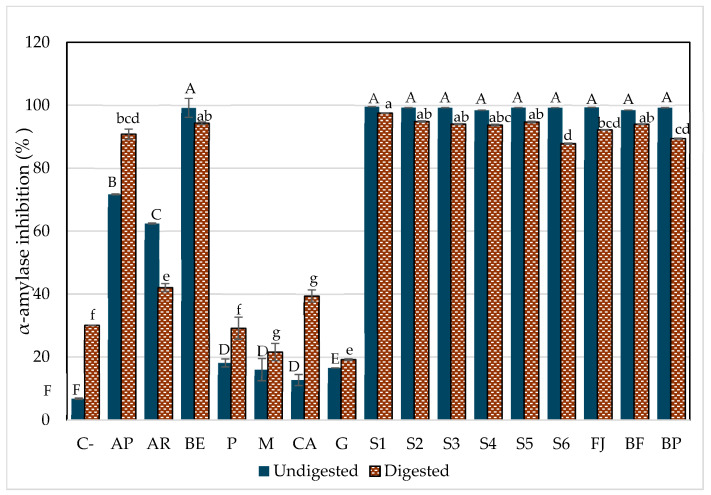
α-Amylase inhibition by the aqueous model systems after in vitro digestion. C-, starch (negative control); AP, pharmaceutical-grade acarbose (1 mg/mL) (positive control); AR, reagent-grade acarbose (1 mg/mL) (positive control); BE, betalain extract (1.7 mg/g); P, pectin (0.9%); M, mucilage (0.2%); CA, citric acid (0.03%); G, glucose (13.7%); S1, BE in water; S2, BE + glucose; S3, BE + citric acid; S4, BE + pectin; S5, BE + mucilage; FJ, fresh juice; BF, formulated beverage; BP, pasteurized beverage. Each bar represents three determinations, while the error bar represents the standard deviation. Uppercase letters indicate statistically significant differences between samples before digestion, and lowercase letters indicate differences between samples after digestion (*p* < 0.05), as determined by Tukey’s test.

**Figure 5 foods-14-01757-f005:**
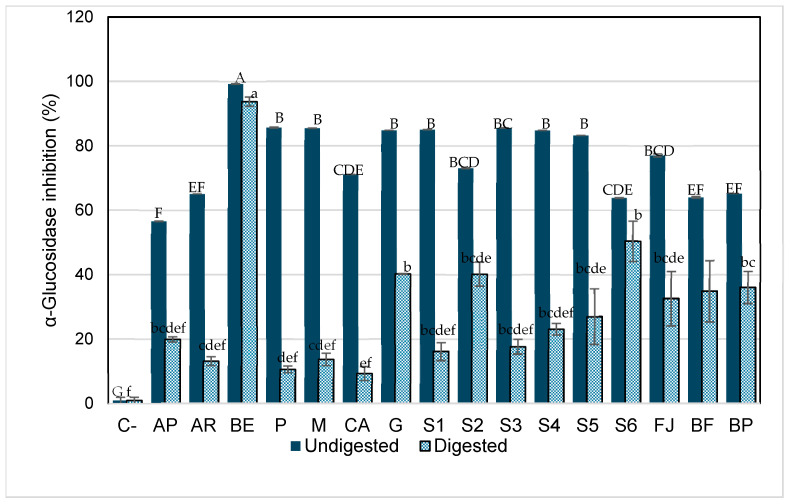
α-Glucosidase inhibition by aqueous model systems after in vitro digestion. C-, starch (negative control); AP, pharmaceutical-grade acarbose (1 mg/mL) (positive control); AR, reagent-grade acarbose (1 mg/mL) (positive control); BE, betalain extract (1.7 mg/g); P, pectin (0.9%); M, mucilage (0.2%); CA, citric acid (0.03%); G, glucose (13.7%); S1, BE in water; S2, BE + glucose; S3, BE + citric acid; S4, BE + mucilage; S5, BE + pectin; FJ, fresh juice; FB, fresh beverage; BP, pasteurized beverage. Each bar is the mean of three determinations while the error bar represents standard deviation. Different letters indicate significantly different values between samples in each phase (*p* < 0.05) according to Tukey’s test.

**Table 1 foods-14-01757-t001:** Aqueous model system (AMS) components and formulation.

AMS	Components		Type	Complexity
**BE**	Betalain extract (1.7 mg/g)		extract	--
**S1**	BE (0.30 mg/mL)	water	mono-component	+
**S2**	BE (0.30 mg/mL)	glucose (13.7%)	bi-component	++
**S3**	BE (0.30 mg/mL)	citric acid (0.03%)	bi-component	++
**S4**	BE (0.30 mg/mL)	pectin (0.9%)	bi-component	++
**S5**	BE (0.30 mg/mL)	mucilage (0.2%)	bi-component	++
**S6**		G+M+P+CA	multi-component	+++
**FJ**	0.30 mg/mL betalains	fresh red prickly pear fruit juice	complex	++++
**BF**	0.30 mg/mL betalains	formulated beverage	complex	++++
**BP**	0.30 mg/mg betalains	formulated beverage, pasteurized	complex	++++

BE—betalain extract, G—glucose, M—mucilage, P—pectin, CA—citric acid, BF—formulated beverage, and BP—pasteurized formulated beverage. Complex food matrix from red prickly pear fruit. All treatments were standardized to a target concentration of 0.30 mg/mL. All samples were standardized to a target concentration of 0.30 mg/mL. Minor variations observed during preparation and quantification were within the expected experimental error range for spectrophotometric methods, and thus were not reported.

**Table 2 foods-14-01757-t002:** Yield of the betalain extract from the Zacatecas variety of red prickly pear juice.

Sample	Betacyanins (mg E_b_/g) λ_max_	Betaxanthins (mg E_I_/g) λ_max_	Total Betalains (mg E_b_+E_I_/g)	Total Betalains in the Sample
Betalain extract (150 g fresh juice)	1.206 ± 0.003	1.018 ± 0.002	2.224 ± 0.005	33.36 ± 0.005
Fresh juice (100 g)	0.144 ± 0.002	0.120± 0.002	0.264 ± 0.005	26.4 ± 0.005

E_b_, betanin equivalents; E_I_, indicaxanthin equivalents. The results are expressed as the means ± standard deviations (n = 3).

## Data Availability

The original contributions presented in the study are included in the article, further inquiries can be directed to the corresponding author.

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
