# Peer review of "Impact of the Food Matrix on the Antioxidant and Hypoglycemic Effects of Betalains from Red Prickly Pear Juice After In Vitro Digestion"

_foods, 2025, doi:10.3390/foods14101757_

Round 1

Reviewer 1 Report

Comments and Suggestions for Authors

Line 9: It is recommended to mention the examination of antioxidative activity along with bioaccessibility and hypoglycemic potential examination since the results were presented in section 3.1. Therefore, consider adding 'antioxidative potential' in the title on Line 248.

Line 14: The oral phase investigation was not described in the Method section – revise.

Line 40: A 'space' is missing before [6].

Line 53: Ensure 'alpha-amylase' and 'alpha-glucosidase' are aligned with the rest of the text.

Line 248: Revise the meaning of 'obtention'.

Line 274: Figure 1 is missing error bars.

Line 282-286: Provide any available data on betalains’ thermal stability. Comment.

Line 287-292: Consider merging this paragraph with the following one.

Line 293: Table 1 contains data already presented in Figure 1. It is suggested to move it to the supplementary material.

Line 333: Please add the source of the structures. Also, consider moving the figure's position after the following paragraph since it is mentioned there for the first time.

Line 335: Use ‘Figure 2’ instead of ‘Figure 3’.

Line 355: Use ‘betalains’ concentration’ instead of ‘betalains’.

Line 394-399: Consider merging the information from this paragraph with the similar one within the first paragraph in section 3.3.1 (Line 345-353).

Line 400: It is recommended to move Figure 3 right after the first paragraph in section 3.3.1 since it is mentioned there for the first time.

Line 462: It is suggested to move Figure 4 right after the second paragraph in section 3.3.3.1 since it is mentioned there for the first time.

Line 559: It is also recommended to mention the examination of antioxidative activity along with bioaccessibility and hypoglycemic potential examination since the results were presented in section 3.1.

Line 595: The title 'Bioaccessibility (%)' can be mentioned only once in Table A1.

Reviewer 2 Report

Comments and Suggestions for Authors

This manuscript presents the authors' investigation of the in vitro hypoglycemic effect of betalain from red prickly pear juice. The manuscript is well designed, rich in data, convincing in conclusion, well expressed in language, and has good academic reference value. On this basis, the author needs to improve some imperfect parts of the manuscript to make it more perfect.

  1. The authors need to add error bars to Figure 1 to show the differences between the data
  2. Table 3 is incorrectly numbered and is mistakenly labeled as Table 1
  3. Page 7, line 283, “This could be due to pasteurization that degrades complex polysaccharides, allowing the release of betalains and improving their interaction with the free radicals.” Can the authors further explain the basis for this conclusion?
  4. The authors spent a lot of space describing the antioxidant capacity of betalain, but lacked a necessary description of linking the antioxidant capacity with the blood sugar-lowering ability (for example, this ability is necessary to judge the stability of betalain). Please add relevant explanations in the introduction or antioxidant section.
  5. Red prickly pear juice contains other ingredients besides betalain, such as other polyphenols, polysaccharides, etc. These ingredients may also play an antioxidant and hypoglycemic role. Can the author briefly explain whether these ingredients will have a significant impact on the effect of betalain?

Reviewer 3 Report

Comments and Suggestions for Authors

The utilization of bioactive compounds from botanicals for the management of diseases such as diabetes mellitus represents a significant area of investigation. This study evaluated the impact of food matrices and thermal processing on the in vitro gastrointestinal bioaccessibility and hypoglycemic potential of betalains in red prickly pear juice. The findings demonstrated that the composite matrix enhances betalain bioaccessibility and augments its hypoglycemic effects. This research provides a foundation for the development of betalain-rich foods for the treatment of type 2 diabetes, expanding the potential applications of red prickly pear.

  1. Strengthen the research on food matrix in other plant extracts or bioactive substances in the introduction.
  2. In the Statistical analysis section (Line 241), specify the level of significance used for statistical testing.
  3. Line 250 should read 333.6, not 336.
  4. Remove "mg" from Table 2 after the values 333.6 and 26.4.
  5. In Figure 1, include standard deviation error bars, correct any missing letters, and place ABTS and DPPH on the y-axis.
  6. Line 293,here are Table 2. Furthermore, a significant portion of the data in Table 2 duplicates that presented in Figure 1. Consider retaining either Figure 1 or Table 2. The results of the significance tests in Table 2 are inconsistent with those in Figure 1; the authors should re-evaluate these. To emphasize the changes in antioxidant capacity before and after digestion, the authors should perform a significance analysis on the data before and after digestion, rather than presenting the absolute differences in the data.
  7. Why do the BE contents in the different treatment groups in Table 1 vary (0.30-0.32 mg/mL)?
  8. Verify the results of the significance tests in Figures 3, 4, and 5.
  9. Carefully review the formatting of the references and provide accurate information. For example, the reference numbers for references 2 and 3 (as page numbers are missing), and ensure consistency in the abbreviation of journal titles.
  10. Has the author analyzed the composition of active substances in red prickly pear juice and their changes after digestion.

Reviewer 4 Report

Comments and Suggestions for Authors

This study explores how different food matrices affect the bioaccessibility and hypoglycemic potential of betalains from red prickly pear juice during in vitro digestion. It found that complex matrices significantly improve betalains stability and bioaccessibility, with mucilage and pectin offering the best protection. Pasteurization reduced stability, and citric acid had a destabilizing effect. Importantly, the juice showed strong inhibition of enzymes linked to glucose absorption, even outperforming acarbose.

This manuscript appears to have scientific high value because the findings are highly applicable to the development of functional beverages or supplements aimed at glycemic control, making it valuable to both academic and industrial sectors.

However, there are some suggestions and revisions that authors can consider for the improvement:

Title

Suggestion: Impact of the food matrix on the antioxidant potential and hypoglycemic effect of bioaccessible betalains from red prickly pear juice after in vitro digestion

Abstract

Line 10: spp, not italic
Line 13-14: .. (BF), and a pasteurized formulated beverage (BP).

Introduction

Line 41: …have shown promising outcome in inhibiting…

Line 52: …glucose absorption through the inhibition on intestinal…

Line 57-58: …functional foods are readily bioaccessible and able to exert…

Line 65: spp, not italic

Line 81: blood glucose level

Materials and Methods

Line 102: 2.2.2

Line 133: remove “and compared against it”

Line 137, 141: pasteurized beverage (BP).

Table 1: pasteurized formulated beverage

Line 144: BP

Line 173, 177: spell out 1 ml

Line 189, 221: the alignment of the equation should be revised.

Line 190: The final volume was further subjected to enzymatic inhibition tests using two digestive enzymes in the small intestines.

Line 194:  …were used as comparison to find out…

Line 195: …if they possess any synergistic effect with…

Line 203: Inhibitory effect on α-amylase was determined…

Line 218: The values obtained…

Line 218: …independent experiments. The enzymatic inhibition percentage was calculated as shown in equation 3…

Line 222: AbsControl: Maximum absorbance..

Line 229: The samples were diluted in 1:8 ratio, equivalent 0.07 mg/ml of betalains.

Line 231: …of α-glucosidase…

Line 239: no equation stated.

Line 241: 2.7 Statistical analysis

Results and Discussion

Line 249: remove eco-friendly

Line 252: remove as corroborated

Line 257: remove DPPH and ABTS assay

Line 264: …lower scavenging potential…

Line 265: remove “ and against the two radicals”

Line 268-269: …juice matrix affected and reduced…, their antioxidant potential was lower.

Line 271: …confirmed… the food matrix were necessary…

Figure 1: the alignment of the figure should be revised.

Line 276: should be Figure 1, not Figure 2

Line 280, 281, 295, 406, 470, 517: p< 0.05

Line 289-290: HAT/PCET could be the possible…

Line 293: Table 3

Line 297: To scavenge the ABTS radical,…

Line 301: …stabilize betalains occurred, as observed in samples…

Line 302: , thus limiting the free functional groups of betalains that…

Line 303: ..the antioxidant function by scavenging the radical.

Line 306: …because the DPPH radical scavenging mechanism is through…

Line 307-309: …aromatic rings, so an ionization potential (IP) and an energy transfer enthalpy (ETE) are required,…

Line 320: …can interact well with…

Line 322: remove “the interactions are increased”

Line 335: Figure 2

Line 341: citation is missing.

Line 373: a higher percentage

Line 374-381: kindly reconstruct this sentence, it is a long sentence with different information

Figure 3, Figure 4, Figure 5: the alignment of the figure should be revised.

Line 417: remove “further validating… food matrix”

Line 434: 3.3.2.1

Line 455: …between 87 and 98% α-amylase activity…

Line 463: …inhibition by aqueous mode systems…

Line 468, 516: three determinations

Line 473: This was…

Line 475: was evident

Line 476: suggested

Line 481: To date,

Line 482-483: …binding and Cl…

Line 484: the activation on enzymatic… active site, consists of three…

Line 491-492: This inhibition likely occurs due to the affinity of the catalytic site of betalains to other compounds…

Line 502: …broken down by enzymes…

Line 503-509: kindly reconstruct this sentence, it is a long sentence with different information

Line 511: …inhibition by aqueous mode systems…

Line 528-532: a citation needed

Line 533-538: a citation needed, and include a figure to show the structure

Conclusion

Kindly include the outcomes of antioxidant in the conclusion.

Round 2

Reviewer 2 Report

Comments and Suggestions for Authors

The author has made comprehensive improvements to the previous comments on the unclear explanation of the purpose of the antioxidant experiment, the flaws in the production of figures and tables, and the unclear presentation of the results of the paper. The manuscript can now fully present the author's intention, that is, to explore the antioxidant and hypoglycemic effects of betalains in red prickly pear juice.
I think the manuscript has met the acceptance criteria in terms of content, and I suggest that the editorial team check whether there are any problems with the language expression before making a final decision.

Reviewer 3 Report

Comments and Suggestions for Authors

The authors have carefully revised the manuscript based on the previous draft in accordance with the reviewers' comments. After the revisions, the paper already meets the requirements of the journal Foods regarding academic integrity and rigor. Additionally, the reviewers have the following suggestions that they hope the authors will consider.

  1. Data analysis methods should be a separate subsection, just like Sections 2.1 to 2.6, and should be assigned a number and a title.
  2. The wavelength used for the determination of betacyanins and betaxanthinsL in 121 and L263 varied. Please check and make them in consistence. Moreover, as the determination method of them have been described in the method, there’s no need to refer the wavelength used in the result.
  3. Ido not think it is necessary to state how much fresh juice was used to extract Betalains, nor the calculation of the total betalains in the sample as listed in Table 2. On the contrary, it is more statistically meaningful for the authors to characterize the extraction rate or the recovery after purification rather than the above-mentioned results.
  4. The authors have not referred to the results in the third row of Table 2 in the main text. I suppose that they are the betalains in the fresh juice before purification. The authors should cite these result in the main text before comparison was made between samples before and after purification in the discussion,  
  5. I suggest that the authors modify the way the BE contents are expressed in Table 1. Given that the slight differences stem from minor variations during sample preparation and quantification, and these small deviations fall within the acceptable experimental error range for spectrophotometric measurement, and since all treatments were standardized to a target concentration of 0.30 mg/mL, the target concentration could be presented in the table, accompanied by a note below to avoidmisleading the readers.
